# Towards Feasible Capsule Network for Vision Tasks

**Dang Thanh Vu † , Le Bao Thai An †, Jin Young Kim and Gwang Hyun Yu ***

Department of ICT Convergence System Engineering, Chonnam National University, 77, Yongbong-Ro, Buk-Gu, Gwangju 61186, Republic of Korea
* Correspondence: sayney1004@gmail.com
† These authors contributed equally to this work.

**Abstract:** Capsule networks exhibit the potential to enhance computer vision tasks through their utilization of equivariance for capturing spatial relationships. However, the broader adoption of these networks has been impeded by the computational complexity of their routing mechanism and shallow backbone model. To address these challenges, this paper introduces an innovative hybrid architecture that seamlessly integrates a pretrained backbone model with a task-specific capsule head (CapsHead). Our methodology is extensively evaluated across a range of classification and segmentation tasks, encompassing diverse datasets. The empirical findings robustly underscore the efficacy and practical feasibility of our proposed approach in real-world vision applications. Notably, our approach yields substantial 3.45% and 6.24% enhancement in linear evaluation on the CIFAR10 dataset and segmentation on the VOC2012 dataset, respectively, compared to baselines that do not incorporate the capsule head. This research offers a noteworthy contribution by not only advancing the application of capsule networks, but also mitigating their computational complexities. The results substantiate the feasibility of our hybrid architecture, thereby paving the way for a wider integration of capsule networks into various computer vision tasks.

**Keywords:** capsule network; computer vision; equivariance; segmentation; pre-trained model

## 1. Introduction

**State-of-the-art Networks:** Convolutional neural networks (CNNs) have dominated the field of computer vision for a decade, solving a wide range of tasks such as image classification [1,2] and segmentation [3,4]. Their success can be attributed to several factors. Firstly, CNNs leverage the translational invariance inductive bias [5], making them adept at recognizing objects under different translations. Secondly, extensive research has focused on developing efficient, high-performance CNN architectures [6–9]. Additionally, the availability of pre-trained models has facilitated the practical adoption of CNNs in both research and enterprise applications. While transformer models, with their self-attention mechanism [10], have gained attention for their flexibility, it is worth noting that they often require substantial amounts of data and computational resources for effective training [11]. Despite this, CNNs remain the preferred choice for smaller datasets due to their ability to quickly adapt to new experiments.

**Invariance vs. Equivariance:** However, CNN models possess inherent limitations that impede their effectiveness in recognizing features across different orientations and encoding comprehensive spatial relationships. The pooling operations employed in CNNs discard essential pose information, resulting in the loss of internal properties such as shape, location, pose, and orientation [12,13]. Furthermore, CNNs rely on large, labeled datasets, face challenges in encoding deformation information, and are vulnerable to unpredictable shifts during testing. Moreover, CNNs tend to rely on memorization rather than comprehension, lacking the ability to capture intricate feature relationships. In contrast, capsule networks (CapsNets) [14] overcome these limitations by incorporating pose, relationships,

and robustness to affine transformations. CapsNets built on the concept of inverse graphics [15] offer superior explanations compared to CNNs. CapsNets provide benefits such as reduced parameter requirements, improved resilience against adversarial attacks [16–18], enhanced interpretability [19], and the capability to handle affine transformation [20–22].

**Challenges:** Despite the promising advancements in CapsNets, several challenges that need to be addressed remain. Training CapsNets continues to pose difficulties, as they have yet to surpass the state-of-the-art benchmarks in a fair manner [20,23–26]. One notable concern is the computational complexity associated with CapsNets, particularly due to the routing algorithm, which necessitates additional memory space and leads to a multiplicative increase in the number of computations, even for relatively small input sizes. Another limitation lies in the specific structure of CapsNets, which inherently limits their applicability primarily to vision-related tasks [27]. While the routing-by-agreement mechanism [28] is strong in theory, its unsupervised clustering nature may not hold up well under heavy input noise. Moreover, unoptimized implementations and a lack of comprehensive architectural understanding further impede researchers from fully harnessing the potential of capsule networks. Additionally, CapsNets struggle with the absence of a pooling mechanism commonly found in traditional CNNs, which results in attempts to account for all aspects of an image, including background noise [29]. Addressing these challenges and developing further improvements will be key to enhancing the robustness and efficiency of capsule networks.

**Contributions:** This study is motivated by the observation that most existing studies in the field of CapsNets have predominantly utilized basic architecture comprising a single convolutional layer for low-level feature extraction, a primary capsule layer for mapping from a feature space to the capsule space, and a class capsule layer for routing to predict class activation and corresponding pose vectors [30–33]. However, this limited depth with only two-to-three capsule layers inherently restricts the expressivity and potential of CapsNets. In prior studies, it has been observed that incorporating dense connections [34] and residual blocks [35,36] can be beneficial for CapsNets. However, these studies have not fully extrapolated the idea due to a lack of comprehensive comparisons with the original models. Additionally, some researchers have attempted to stack capsule layers by simplifying the routing mechanisms [16,24,31,37–40], but this approach often results in a non-faithful implementation of CapsNets. Building upon these insights, our study aims to bridge these gaps by thoroughly investigating the deeper backbone within the context of CapsNets. Furthermore, we strive to provide a faithful and comprehensive evaluation by conducting thorough comparisons with the original models. Our contributions are as follows:

- We explore deeper architectures to unlock the capabilities of CapsNets;
- By leveraging the strengths of various backbone models, we propose a capsule head wrapping (CapsHead) approach and carefully experiment with modifications to the capsule head and routing mechanism;
- We aim to enhance the expressivity and performance of CapsHead in tasks such as classification, medical image segmentation, and semantic segmentation.

This paper follows a structured organization: Section 2 presents the methodology, then Section 3 reviews related works, Section 4 showcases experimental results, and Section 5 concludes the study.

## 2. Related Works

**Capsule network in vision tasks**: Capsule networks have made significant strides in various vision tasks across domains. They have been successfully applied to hyperspectral image classification as spectral–spatial units [41]. In remote sensing image scene classification, CNN–CapsNet architectures have achieved enhanced results on diverse datasets [42]. For action detection in videos, VideoCapsuleNet, a 3D capsule network, has demonstrated superior performance with 3D capsule convolution [43]. In image synthesis, CapsuleGAN and CapsGAN have excelled at modeling image data distribution and capturing geometric



transformations [44,45]. Adversarial attacks on CapsHeads have been investigated, with studies such as CapsAttacks and Capsule-FORENSICS focusing on vulnerability analysis and detecting forged images and videos [17,18]. In the medical imaging domain, CapsNets have shown promise in lung cancer screening and medical image segmentation [38,46]. Furthermore, CapsNets have contributed to advancements in brain tumor type classification, offering improved interpretability of learned features [47]. These advancements collectively demonstrate the potential and versatility of CapsNets in computer vision applications.

**Routing mechanism**: Studies have explored various routing mechanisms in CapsNets. Dynamic routing enables capsules to route information based on agreement scores [14], while matrix capsules with EM routing use an iterative expectation–maximization process for routing coefficients [28]. Alternative methods include inverted dot-product attention routing [48], shortcut routing for reduced complexity [21], and self-routing capsule networks with subordinate networks [16]. Generalized capsule networks train coupling coefficients for flexibility [32]. Efficient routing algorithms like self-attention routing and weighted kernel density estimation improve speed and efficiency [22,49]. Max–min normalization allows for independent values within bounds [50], and spectral capsule networks employ singular value decomposition for routing [51]. These studies advance routing mechanisms, addressing computational efficiency, attention-based routing, and coefficient optimization.

**Deep capsule network**: Numerous research efforts have focused on enhancing CapsNets for complex data analysis. Investigations into model modifications, dynamic routing algorithms, and the use of convolutional capsule layers (Conv-Caps Layers) have been proposed to improve performance [39,52,53]. Dense capsule networks (DCNets) and diverse capsule networks (DCNets++) replace standard convolutional layers with densely connected convolutions [34], and DeepCaps introduces a novel 3D convolution-based dynamic routing algorithm [52]. The diverse enhanced capsule network (DE-CapsNet) leverages the advantages of DCNet++ and employs residual convolutional layers for diverse enhanced primary capsules [35]. Neural network encapsulation approximates the routing process with master and aide branches, while CapsNets with residual connections introduce skip connections to facilitate training depth [24,36]. These studies collectively advance CapsNets for complex data analysis, augmenting their efficacy in various applications.

## 3. Methods

### 3.1. Preliminaries

**Capsule Network**: In the context of CapsNets, the fundamental idea is to introduce capsules that encapsulate pose information along with other instantiation parameters, such as color and texture, for different parts or fragments of an object. This structure is characterized by being deep in width rather than height, resembling a parse tree [13] where each active capsule selects a parent capsule in the next layer. The underlying principle is that as the viewpoint of an object changes, the corresponding pose matrices should be coordinated to maintain the voting agreement.

A CapsNet typically consists of three key components: a stack of convolutional layers responsible for extracting features, a primary capsule layer that transforms these features into capsule representations, and a stack of capsule layers that incorporate the routing mechanism. In CapsNets, capsules in a lower layer $L_i$ (children) are routed to capsules in a higher layer $L_j$ (parents), creating a connection between the two layers. In every layer, there are multiple capsules, each characterized by an instantiation parameter pose vector $S_i \in \mathbb{R}^P$ [14] or matrix $S_i \in \mathbb{R}^{P \times P}$, and an activation probability $a_i$ [28]. The pose in each capsule encodes the relationship of an entity to the viewer, while the activation probability represents its presence. Using its pose matrix, $S_i$, each lower-level capsule contributes a vote to determine the pose of a higher-level capsule. This is achieved by multiplying the pose with a trainable viewpoint-invariant transformation weight matrix.

$$V_{j|i} = S_i W_{ij} \tag{1}$$

In essence, the trainable weights, $W_{ij}$, enable the capsules to learn affine transformations, allowing them to capture and represent the part–whole relations within the data.

**Routing Methods**: Routing-by-agreement is a dynamic information flow through the network by determining connections between successive layers of capsules at runtime. Unlike traditional neural networks, where cross-layer connections are determined solely by network parameters, routing enables the adjustment of magnitudes and relevance from lower capsules to higher capsules, ensuring the activation of relevant higher-level counterparts and effective transmission of pattern information. The concept of routing can be likened to clustering logic [28,49] where higher-level parent capsules receive votes from multiple lower-level child capsules within their receptive fields. However, capsule routing differs from regular clustering as each cluster has its own learnable viewpoint-invariant transformation matrix, enabling a unique perspective on the data and facilitating faster convergence by breaking symmetry. Different approaches to routing have been explored in CapsNets [16,21,22,32,49,50,54].

The pre-activation of a capsule, $j$, is computed as the sum of the association coefficients, $c_{ij}$, multiplied by the predictions, $V_{j|i}$, of the lower-level capsules $i$.

$$S_j = \sum_i c_{ij} V_{j|i} \tag{2}$$

These association coefficients are determined through iterative routings, with examples outlined below.

1.  Dynamic routing [14]: The agreement is measured by cosine similarity, and the coupling coefficients are updated as follows:

$$
\begin{aligned}
c_{ij} &\leftarrow \frac{e^{b_{ij}}}{\sum_j e^{b_{ij}}} \\
b_{ij} &\leftarrow b_{ij} + \left\langle S_j, V_{j|i} \right\rangle
\end{aligned}
\tag{3}
$$

2.  EM routing [28]: This refers to using an EM algorithm to determine the coupling as a mixture coefficient with cluster assumption that the votes are distributed around a parent capsule.

$$
\begin{aligned}
c_{ij} &\leftarrow \frac{a_j p_j}{\sum_j a_j p_j} \\
p_j &\leftarrow \frac{1}{\sqrt{\prod_h 2\pi \left(\sigma_j^h\right)^2}} \exp\left(-\sum_h \frac{\left(V_{j|i}^h - S_j^h\right)^2}{2\left(\sigma_j^h\right)^2}\right)
\end{aligned}
\tag{4}
$$

Activation of a parent capsule, $j$, occurs when there is a substantial consensus among the votes with the parent capsule. This consensus leads to the formation of a compact cluster ($\sigma_j \to 0$) in the $D$-dimensional space.

3.  Max–min routing [50]: Instead of using SoftMax, which limits the range of routing coefficients and results in mostly uniform probabilities, this study proposes the utilization of max–min normalization. Max–min normalization ensures a scale-invariant approach to normalize the logits.

$$
\begin{aligned}
c_{ij} &\leftarrow \frac{b_{ij} - \min_j b_i}{\max_j b_i - \min_j b_i} \\
b_{ij} &\leftarrow b_{ij} + \left\langle S_j, V_{j|i} \right\rangle
\end{aligned}
\tag{5}
$$

4.  Fuzzy routing [21]: To address the computational complexity of EM routing, Vu et al. introduce a routing method based on fuzzy clustering, where the coupling between capsules is represented by fuzzy coefficients. This approach offers a more efficient

alternative to EM routing, reducing the computational demands, while still enabling effective information flow between capsules.

$$c_{ij} \leftarrow \frac{w_{ij}^m}{\sum_k w_{kj}^m}$$
$$w_{ij} = \frac{1}{\sum_k \left( \frac{\|V_{j|i} - S_j\|}{\|V_{k|i} - S_k\|} \right)^{\frac{2}{m-1}}} \tag{6}$$

In the Fuzzy C-means algorithm, the parameter $m$ controls the degree of fuzziness in the clustering process. Larger values of $m$ lead to fuzzier clusters, where the membership values $w_{ij}$ can approach either 0 or 1. When $m$ is set to 2, the objective of Fuzzy C-means aligns with that of the traditional K-means algorithm.

Routing in CapsNets filters out contributions from submodules with noisy information, suppresses output capsules with high variance predictions, and promotes consensus among capsules. However, the assumption of spherical or normal distribution of prediction vectors may not hold in real-world data with variability and noise. It also has inherent weaknesses, the unsupervised clustering nature of routing requires repeated computations, increasing computational complexity compared to one-pass feed-forward CNNs.

### 3.2. Hybrid-Architecture Capsule Head

Our proposed architecture incorporates several design elements, with a focus on utilizing a backbone model and stacking a task-driven capsule head over the extracted feature maps. We present various design demonstrations, showcasing different configurations for the integration of CapsHead within the architecture, shown in Figure 1.

(1) In the first design, we add adaptive average pooling to reduce the feature maps' dimension and a fully connected capsule layer. This configuration enables the transformation of backbone features into capsules through the primary caps layer, followed by routing through the FCCaps layer.

(2) In the second design, we again employ average pooling after the backbone feature extraction, followed by a projection operation to enhance the capacity of the embedding space. Then, we split by channel dimension to aggregate the capsules. Subsequently, routing is applied to these capsules to get the next-layer capsules.

(3) In the third design, we remove the average pooling, but keep the projection and channel splitting. By adopting these modifications, the routing layer can effectively capture spatial information, making it well-suited for segmentation tasks. However, for classification tasks, we extend the functionality by incorporating capsule pooling, which allows us to reduce the number of class capsules to the desired target.

(4) Lastly, the fourth design directly explores the splitting of feature maps, followed by projection and adaptive capsule routing. This configuration enables a more adaptive and flexible routing mechanism based on the spatial characteristics of the feature maps.

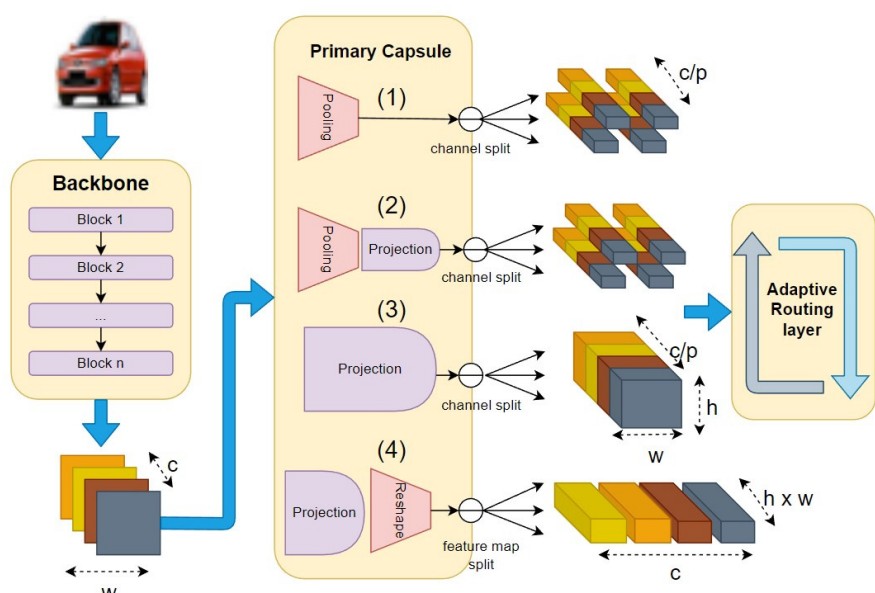

**Figure 1.** Proposed architecture with four configurations. The architecture comprises three key components: a backbone network, a primary capsule layer, and a capsule routing layer. We focus on modifying the primary capsule layer by introducing different settings, such as non-linear projection, and toggling between channel split and feature map split.

These design demonstrations illustrate the versatility and flexibility of our proposed architecture, showcasing different configurations for integrating CapsHead within the backbone model. Each design offers unique advantages and possibilities for improving the performance and capabilities of capsules in various vision tasks.

Our designed architectures are developed based on three key criteria which allow us to tailor the CapsNets to a specific task and size of dataset.

- Firstly, we consider whether the feature maps extracted from the backbone model are before or after the pooling layer. For one-dimensional feature maps, after the pooling layer, which represent high-level features condensed into a single vector, they can be directly used for linear evaluation and analysis. On the other hand, two-dimensional feature maps, before the pooling layer, capture rich contextual information, particularly beneficial for interpreting the entire model or visualizing the learned features.
- The second criterion pertains to the interpretation of capsules. Capsules can be seen as encapsulating either channels or feature maps. In the channel-based interpretation, a capsule represents a pose vector constructed at a specific 1-pixel location, with the channel dimension serving as the capsule pose. The total number of capsules is determined by the number of pixel locations. Alternatively, in the feature map-based interpretation, each feature map constitutes a capsule, and we utilize average adaptive pooling to obtain the desired dimension of the capsule pose. In this case, the channel size corresponds to the number of capsules.
- Lastly, we consider the mapping of feature vectors to the primary capsule space. We provide the flexibility of either directly using the feature space spanned by the backbone model or incorporating a non-linear projection head to map the feature vectors to the primary capsule space. This allows for a more tailored and optimized representation of capsules. In this study, we craft the projection head using a multi-layer perceptron with two-to-three layers, incorporating non-linear activation functions like ReLU.

In our proposed architecture, we introduce an adaptive routing layer that facilitates the routing of capsules to class-level capsules, regardless of the size of the feature maps. This idea is inspired by the concept of capsule pooling introduced in [16,52]. If the size of the feature maps is different from $1 \times 1$, we first perform routing to capture the spatial

relationships, and then apply capsule pooling to achieve the desired output size, which is typically $1 \times 1$, and flatten its capsule activations which are used for the classification task.

The proposed architecture combines a backbone model with CapsHeads to leverage their respective strengths and enhance the overall performance. The backbone model [1–4,11], which can be a pre-trained deep neural network, serves as a feature extractor, capturing high-level features from the input data. These features are then fed into the CapsHead, which introduce capsule layers to capture spatial relationships and enable richer representations. The combination of the backbone model and CapsHead offers several advantages. Firstly, the backbone model provides a strong foundation of feature extraction, leveraging its ability to learn complex patterns and representations from large-scale datasets. This allows our model to benefit from the informative features extracted by the backbone model, enhancing their discriminative power. Additionally, capsules encapsulate pose information and activation, allowing them to capture spatial relationships between features. This makes them well-suited for tasks requiring the understanding of object orientation, pose, and spatial arrangements.

## 4. Experiments

### 4.1. Dataset

We considered four datasets.

- CIFAR10: CIFAR10 is an image classification dataset that contains a total of $60,000$ images. It consists of 10 different classes, with 6000 images per class. Each image is a $32 \times 32$ color image, making it a widely used benchmark for evaluating image classification algorithms.
- CIFAR100: CIFAR100 is an extension of CIFAR10, offering more fine-grained labeling. It comprises a total of $60,000$ images across 100 classes, with 600 images per class. This dataset provides a challenging task for fine-grained image classification, enabling researchers to evaluate algorithms with increased specificity.
- LungCT-Scan: The LungCT-Scan dataset is designed specifically for lung image analysis in medical imaging research. It consists of computed tomography (CT) scan images of the lungs. The purpose of the dataset is for image segmentation. We used 213 images for training and 54 images for validation.
- VOC-2012: VOC-2012, or the PASCAL VOC dataset, is a benchmark dataset for object detection, segmentation, and classification. It consists of approximately $11,540$ images in total. The dataset includes annotations such as object bounding boxes and pixel-level segmentation for various object categories. In this study, we use 1464 images for training and 1449 images for validation.

### 4.2. Configurations

All experiments used the same settings, where the cross-entropy loss was utilized. The metric used for evaluation was accuracy. The training was conducted for 100 epochs with a batch size of 64. The input images were resized to a size of $224 \times 224$ pixels. The training was performed on two Titan V GPUs for the classification task and two 3090 GPUs for the segmentation task. The optimizer used was Adam, and a step-based learning rate scheduler was employed with an initial learning rate of 0.001. The learning rate was reduced by a factor of 0.8 every 5 epochs. Early stopping was applied to prevent overfitting during training. Implementation in the PyTorch framework is available at https://github.com/Ka0Ri/Capsule-Network (accessed on 4 September 2023).

In the CapsHead architecture, we maintained a consistent dimension of 4 for the capsules, performed 3 routing iterations, and incorporated a projection head with a hidden dimension of 512, resulting in 32 capsule styles. During fine-tuning, we utilized a pretrained model from the ImageNet1K dataset provided by the PyTorch hub and enabled gradient updates for all layers. Additional specific configurations for each experiment will be elaborated upon in their respective sections.

*4.3. Results*

4.3.1. Linear Evaluation of the Classification Task

In the context of linear evaluation experiments, the primary objective is to provide evidence supporting the superiority of the capsule head as a classifier over the conventional fully connected (FC) layer. This advantage can be attributed to the unique "Routing Mechanism" employed in CapsNet. To initiate the evaluation process, we first extract feature maps from a pre-trained ResNet18 model. These feature maps are obtained from the output of the fourth block, situated just before the average pooling layer. It is worth noting that this specific feature map is instrumental in encapsulating all relevant information from preceding receptive fields, making it a valuable tool for model interpretation. Consequently, our evaluation framework comprises two key inputs for our model: before average pooling, dimensions of $512 \times 7 \times 7$ because the input size is $224 \times 224$, and feature vector after average pooling with dimensions of 512.

For the CIFAR10 dataset, the performance of CapsHeads with dynamic routing and settings (3), using the max–in normalization technique, achieved an accuracy of 89.18%. This outperformed the baseline classifiers, such as support vector machine (SVM), with an accuracy of 89.2%, and fully connected (FC) networks with two hidden layers and one layer, which achieved accuracies of 89.1% and 87.97%, respectively. Similarly, for the CIFAR100 dataset, the CapsHeads with dynamic routing and settings (3) attained an accuracy of 69.47%, surpassing the SVM classifier with an accuracy of 68.02%, and the FC networks with two hidden layers and one layer, achieving accuracies of 66.82% and 66.43%, respectively. The results in Table 1 demonstrate that CapsHeads with setting 3 perform competitively against traditional classifiers, such as SVM and FC networks, on both CIFAR10 and CIFAR100 datasets. The CapsHeads approach leverages the benefits of capsule vector in capturing intricate spatial relationships, and the utilization of routing with specific settings further enhances their classification performance.

**Table 1.** Linear evaluation performance of CapsHeads (dynamic routing with setting (3) and no projection) compared to baseline models on the classification task.

| Dataset | Classifier | Settings | Accuracy |
|---------|-----------|----------|----------|
| CIFAR 10 | SVM | | 89.2 |
| | FC | 2 hidden layers | 89.1 |
| | FC | 1 layer | 87.97 |
| | CapsHead | Max-Min, setting (3) | **89.18** |
| CIFAR 100 | SVM | | 68.02 |
| | FC | 2 hidden layers | 66.82 |
| | FC | 1 layer | 66.43 |
| | CapsHead | Dynamic, setting (3) | **69.47** |

4.3.2. Performance on Segmentation Task

In this study, we aim to test the efficacy of the capsule head in solving segmentation tasks, specifically in the third setting. Our design is optimized to preserve the spatial size required for making predictions at each pixel, making it suitable for segmentation.

Table 2 provides a comprehensive comparison between CapsHead (setting (3) with dynamic routing and no projection) and traditional CNN baselines (with one hidden layer) on the segmentation task for two distinct datasets: CT-Lung-Scan and VOC-2012. For the CT-Lung-Scan dataset, CapsHead demonstrates remarkable performance, achieving a Dice score of 97.59, surpassing the results of the CNN baselines. Specifically, the FCN model achieves a Dice score of 96.06, while Deeplab achieves 97.69. Similarly, on the VOC-2012 dataset, CapsHead showcases its proficiency by achieving a Dice score of 86.58. In comparison, the FCN baseline only manages a Dice score of 80.24, and Deeplab obtains 80.27. These findings further validate the efficacy of CapsHead in segmentation tasks and illustrate their ability to outperform traditional CNN models.

**Table 2.** Performance of CapsHead compared to the CNN baselines on segmentation tasks.

| Dataset | Backbone | Head | Dicescore |
|---|---|---|---|
| CT-Lung-Scan | FCN | CNN | 96.06 |
| | Deeplab | | 97.69 |
| | FCN | CapsHead | **97.59** |
| | Deeplab | | **97.70** |
| VOC 2012 | FCN | CNN | 80.24 |
| | Deeplab | | 80.27 |
| | FCN | CapsHead | **86.58** |
| | Deeplab | | **86.11** |

The results presented in Table 2 underscore the advantages of utilizing the CapsHead architecture to enhance segmentation performance. CapsHead, with its inherent capacity to capture intricate spatial relationships and process hierarchical visual parts in an image, emerges as well-suited for image segmentation tasks, as shown in Figure 2. This finding highlights the tremendous potential of CapsNets in the domain of image segmentation.

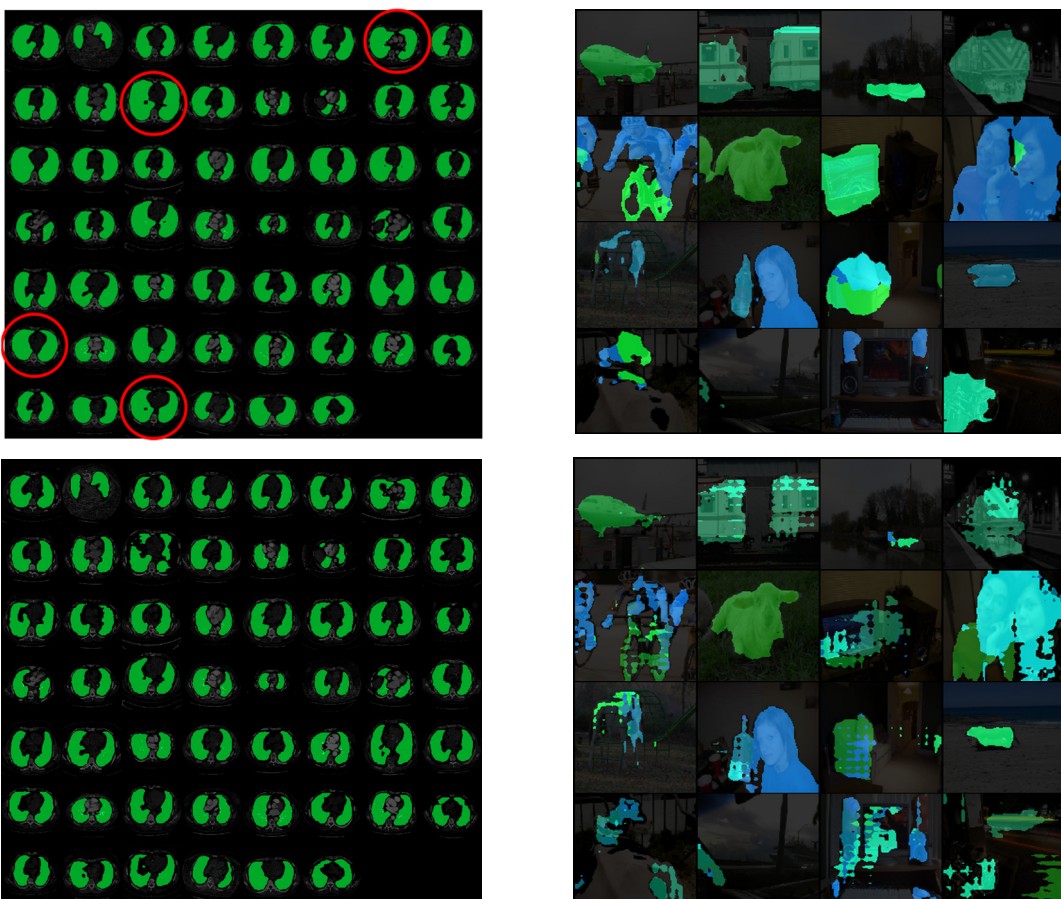

**Figure 2.** Segmentation masks of Lung-CT-Scan (**left**) and VOC2012 (**right**) datasets from: (**first row**) FCN-CapsHead, (**second row**) FCN. Comparing segmentation result in read circles we clearly see that CapsHead produces better overlapping between a mask and ground truth.

### 4.3.3. Pretrained and Fine-Tuned Evaluation

In this study, our main objective is to investigate whether CapsNet can enhance the fine-tuning model in a downstream task. We aim to assess the potential benefits of

incorporating capsule structures in mainstream CNNs. We conduct experiments involving two variations of CapsNet configurations:

- CapsNet with a pre-trained backbone: The pre-trained model serves as the starting point, and we subsequently fine-tune the entire network, including the CapsHead.
- CapsNet without a pre-trained backbone: Here, the entire network, including the capsule structures, is trained from scratch on the target downstream task.

For CIFAR10, when CapsHead is trained from scratch, it achieves an accuracy of 83.7% with ResNet18 and 86.03% with DenseNet. However, with a pre-trained backbone, the performance significantly improves to 94.08% with ResNet18 and 94.97% with DenseNet. Similarly, for CIFAR100, CapsHead with a pre-trained ResNet18 backbone achieves an accuracy of 73.03%, which is notably higher than the 54.29% achieved when trained from scratch. Additionally, CapsHead with a pre-trained DenseNet backbone achieves a significant accuracy of 79.91%. The results in Table 3 indicates that leveraging a pre-trained backbone significantly boosts the performance of CapsHead in both CIFAR10 and CIFAR100 datasets, indicating the importance of transfer learning in enhancing the capabilities of CapsNets for vision tasks. Figure 3 also clearly shows that Pretrained backbones outperform scratch training and DenseNet backbones are better than ResNet backbones. By utilizing the knowledge learned from a large-scale dataset, CapsHead can effectively capture intricate spatial relationships and achieve impressive accuracy for image classification tasks.

**Table 3.** Performance of CapsHead (setting (1), dynamic routing) with a pre-trained backbone and from scratch.

| Dataset | Backbone | Scratch | With Pre-Trained |
|---|---|---|---|
| CIFAR 10 | ResNet18 | 83.7 | 94.08 |
| | DenseNet | 86.03 | **94.97** |
| CIFAR 100 | ResNet18 | 54.29 | 73.03 |
| | DenseNet | 55.08 | **79.91** |

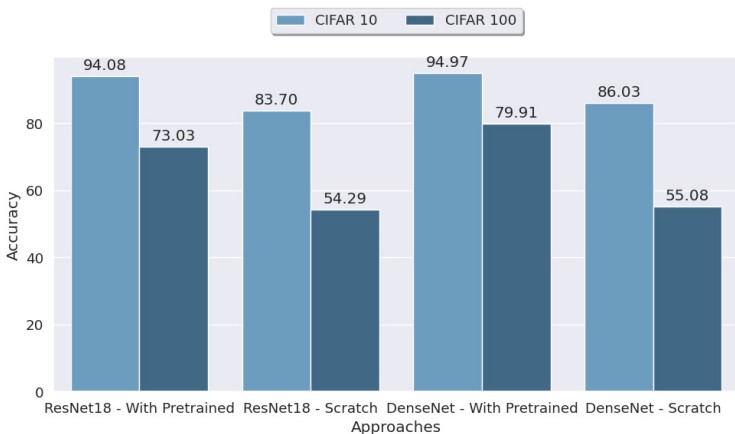

**Figure 3.** Comparison of accuracy for different backbones from scratch and pre-trained on the CIFAR10 and CIFAR100 datasets.

### 4.3.4. Ablation Study

In essence, the strength of the pre-trained backbone model provides a solid foundation, enabling us to concentrate on the hyperparameters specific to our design. In this section, we focus on investigating the type of routing method and the structure of the primary capsule layer, among other parameters, across four different settings. These experiments allow us to gain insights into the impact of these architectural choices on the overall performance of the CapsNet. During this experiment, unless specified otherwise, we adhered to the default setting as follows: the first (1) architecture was utilized, where capsules are directly

constructed from the feature map of the ResNet18 backbone model, and the routing method employed was dynamic routing.

Table 4 presents an ablation study on the capsule head's performance for the classification task using the CIFAR10 dataset. In the primary capsule configuration, four designs are investigated: architecture (1) with a single routing layer with 200K parameters, achieving an accuracy of 83.7%; method (2) with the projection of two hidden layers with 800 K parameters, resulting in an accuracy of 83.21%; method (3) with two hidden layers with 5.1 M parameters and keeping spatial dimension, yielding an accuracy of 84.02%; and (4) capsule styles are constructed from feature maps, with 500 K parameters, achieving the highest accuracy of 84.26%. The accuracy scores for each configuration demonstrate the trade-offs between parameter counts and model performance. Additionally, we evaluate four routing methods: dynamic with an accuracy of 83.7%, max–min with an accuracy of 80.64%, EM with an accuracy of 80.1%, and Fuzzy with an accuracy of 82.18%.

**Table 4.** Ablation study on CapsHead with classification tasks on the CIFAR10 dataset.

| Tunning | Value | Accuracy | Params of Capsule Head |
|---|---|---|---|
| Primary Capsule | (1) | 83.7 | 700 K |
| | (2)—2 hidden layers | 83.21 | 1.5 M |
| | (3)—2 hidden layers | 84.02 | 5.6 M |
| | (4)—2 hidden layers | **84.26** | 1 M |
| Routing Method | Dynamic | **83.7** | |
| | Max–Min | 80.64 | |
| | EM | 80.1 | |
| | Fuzzy | 82.18 | |

Figure 4 illustrates the outcomes of our ablation studies. We observe that there is no distinct trend or significant variation between different primary capsule architectures and routing types. Nevertheless, approaches employing cosine similarity and vector length, such as dynamic and max–min routing, appear to yield better results compared to those utilizing clustering assumptions, such as EM and fuzzy routing. Furthermore, methods (3) and (4), which leverage more spatial information to construct capsules, demonstrate higher effectiveness compared to approaches solely relying on feature vectors. These findings shed light on the strengths and limitations of various capsule configurations and routing techniques, guiding future developments in the CapsNet architecture design.

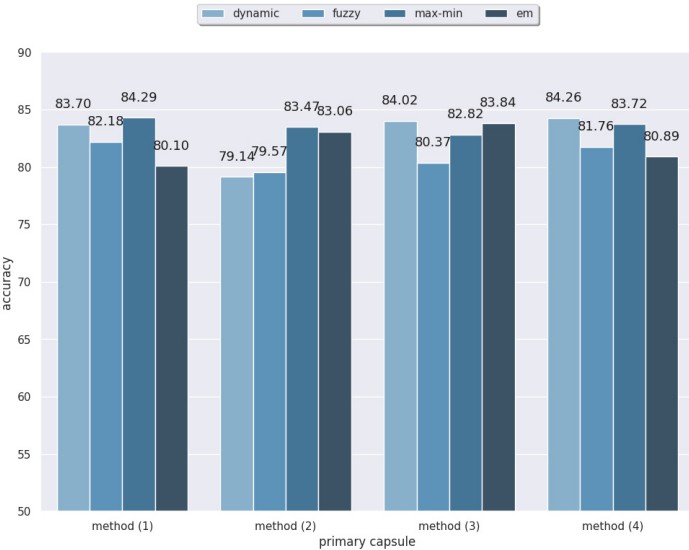

**Figure 4.** CapsHead ablation studies.

Table 5 present a comprehensive comparison of the performance of CapsHead, our proposed method, against several prior works on both the CIFAR10 and CIFAR100 datasets. For the CIFAR10 dataset, the baseline model accuracies range from 73.3% (HitNet) to 95.45% (Encapsulation). Among the various approaches, CapsHead achieves an impressive accuracy of 94.97%, positioning it as a competitive and effective solution for the CIFAR10 classification task. Similarly, on the CIFAR100 dataset, the Encapsulation method serves as the baseline, with an accuracy of 73.33%. In this context, CapsHead demonstrates its capability to achieve a notable accuracy of 79.91%, showing its potential for handling more complex datasets.

**Table 5.** Performance of CapsHead compared to prior works.

| Dataset | Study | Accuracy (%) |
|---------|-------|--------------|
| CIFAR 10 | HitNet [20] | 73.3 |
| | Two-phase routing [37] | 75.82 |
| | KDE Routing [49] | 84.6 |
| | DCNET++ [34] | 89.32 |
| | Self-Routing [16] | 92.14 |
| | DeepCaps [52] | 92.74 |
| | DE-CapsNet [35] | 92.96 |
| | Encapsulation [24] | 95.45 |
| | **CapsHead (ours)** | **94.97** |
| CIFAR 100 | Encapsulation [24] | 73.33 |
| | **CapsHead (ours)** | **79.91** |

The results in Table 5 showcase the superior performance of CapsHead in comparison to several prior works on both the CIFAR10 and CIFAR100 datasets. For CIFAR10, the CapsHead accuracy is only slightly below the state-of-the-art Encapsulation model, emphasizing its robustness and effectiveness in handling image classification tasks. Moreover, CapsHead outperforms all other baseline methods listed on CIFAR10, further solidifying its position as a promising approach. For CIFAR100, CapsHead again shows its ability to excel, surpassing the baseline Encapsulation method. While the difference in accuracy is not as substantial, it is noteworthy that CapsHead maintains its competitiveness across diverse datasets.

### 4.3.5. Limitation

Our study indeed acknowledges several limitations. Firstly, our study's limitations are linked to the scope of experimentation. While we tested our model on the CIFAR10 and VOC2012 datasets, they might not comprehensively represent the diversity and scale of contemporary computer vision challenges. Emerging methods are often evaluated on extensive and varied datasets, and our focus on these specific datasets could impact the generalizability of our model. Additionally, the increasing attention toward transformer-based backbones highlights a limitation in our study, as we predominantly centered on CNN backbones. The shifting landscape of backbone architectures underscores the need for a more comprehensive exploration that encompasses transformer-based architectures. Moreover, while our approach shows promise, we acknowledge that we could not conclusively establish the efficiency of CapsHead with augmented data. Further investigation in this area is necessary. Last is the absence of a robust theoretical exploration of the equivariance property of CapsHead. This theoretical foundation remains a future avenue of inquiry for our work.

### 5. Conclusions

This study introduces a pioneering hybrid architecture for capsule networks, seamlessly integrating a pre-trained backbone model with a dedicated capsule head tailored to the task. Through extensive experimentation, our approach demonstrates superior

performance in a range of vision tasks, delivering particularly notable advancements in pixel-level tasks like segmentation.

However, it is important to acknowledge the limitations of our study. While our hybrid architecture showcases promising results, further optimization is needed to address its computational complexity, which could hinder its scalability to larger and more intricate datasets. Additionally, the trade-offs between performance gains and potential increased training times should be explored in more depth. Furthermore, we did not explicitly demonstrate the equivariance property of the proposed architecture, both theoretically and practically. While the potential for equivariance exists within our framework, a comprehensive theoretical exploration within the representation learning context is a direction we intend to pursue in future studies.

**Author Contributions:** Conceptualization, L.B.T.A.; methodology, L.B.T.A., D.T.V. and J.Y.K.; software, D.T.V.; validation, G.H.Y., L.B.T.A. and D.T.V.; formal analysis, D.T.V.; investigation, L.B.T.A. and G.H.Y.; resources, L.B.T.A.; data curation, G.H.Y.; writing—original draft preparation, L.B.T.A.; writing—review and editing, L.B.T.A. and G.H.Y.; visualization, D.T.V.; supervision, J.Y.K.; project administration, G.H.Y. and J.Y.K.; funding acquisition, G.H.Y., J.Y.K. All authors have read and agreed to the published version of the manuscript.

**Funding:** This work was partly supported by the Institute of Information & Communications Technology Planning & Evaluation (IITP) grant funded by the Korea government (MSIT) (no. 2021-0-02068, Artificial Intelligence Innovation Hub) and the MSIT (Ministry of Science and ICT), Korea, under the Innovative Human Resource Development for Local Intellectualization support program (IITP-2023-RS-2022-00156287) supervised by the IITP (Institute for Information & Communications Technology Planning & Evaluation).

**Institutional Review Board Statement:** Not applicable.

**Informed Consent Statement:** Not applicable.

**Data Availability Statement:** The CIFAR10 and CIFAR100 datasets are published at https://www.cs.toronto.edu/~kriz/cifar.html (accessed on 1 March 2023), the VOC2012 dataset is published at http://host.robots.ox.ac.uk/pascal/VOC (accessed on 1 March 2023), the Lung-CT-Scan dataset can be downloaded from https://www.kaggle.com/code/travishong/image-segmentation-for-lung-ct (accessed on 2 February 2023).

**Acknowledgments:** We extend our heartfelt gratitude to Chi Hwan Hwang and Young Chang Kim at SEO Corporation, Gyeonggi-do, Republic of Korea. Without their invaluable contributions, this study would not have reached its current state. Their roles spanned from managing funding to advising on content. We deeply appreciate their dedication and significant contributions to our work.

**Conflicts of Interest:** The authors declare no conflict of interest.

## Abbreviations

| | |
|---|---|
| CNN(s) | Convolutional neural network(s) |
| CapsNet(s) | Capsule network(s) |
| CapsHead(s) | Capsule head(s) |

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
