# Peer review of "Towards Feasible Capsule Network for Vision Tasks"

_applsci, doi:10.3390/app131810339_

Round 1

Reviewer 1 Report

The work of the manuscript entitled “Towards feasible Capsule Network for vision tasks” seems to have been carefully completed and gave some detailed results. However, the manuscript still should be improved a lot as it contains many points that needs to be revised:

1.       The abstract is too limited. It should generally include the research background and purpose(i.e., what is the research gap?), research methods, research results, research importance and potential impact. The number of words should be controlled to about 200. It should be modified to show the academic contribution and achievement of the manuscript more clearly.

2.       The authors should increase the number of keywords. It is appropriate to arrange from 5 keywords to 7 keywords.

3.       Abbreviation is needed before the Introduction section.

4.       It is strongly recommended that the references should be renewed, most of the references should be published within 5 years, as the domain of CV develops rapidly.

5.       Introduction must be enriched by recent published articles; I could recommend you to read and integrated the following articles: 10.7717/peerj-cs.1400; https://doi.org/10.1007/s44196-023-00233-6; 10.2352/J.ImagingSci.Technol.2023.67.3.030402; 10.7717/peerj-cs.353; 10.1109/ACCESS.2021.3074937; https://doi.org/10.1016/j.patcog.2021.108153

6.       It would be better to place the section of related work after the introduction.

7.       Some figures are not clear, it is hard to see the words in them and the resolution should be greatly improved.

8.       Conclusion is too limited. The authors may give the details of their paper's novelty with short descriptions.

Moderate editing of English language required. Please have someone competent in the English language and the subject matter of your paper go over the paper and correct it.

Author Response

Thank you for the opportunity to revise our manuscript, "Towards feasible Capsule Network for vision tasks". We appreciate the careful review and constructive suggestions. We believe that the manuscript is substantially improved after making the suggested revisions. Following this letter are the editor and reviewer comments with our responses in italics, including how and where the text was modified. Changes made in the manuscript are marked using comments. The revision has been developed in consultation with all coauthors, and each author has approved the final form of this revision. The agreement form signed by each author remains valid. Thank you for your consideration.  

Sincerely, 

Dang Thanh Vu 

Reviewer 2 Report

This manuscript presents a novel architecture called CapsHead based on capsule networks. The architecture is presented based on criteria that can adapt to produce optimal performance based on the task objective(s) and dataset size(s). The authors also present four design modifications to the architecture that claim to showcase versatility and flexibility in performing tasks when integrated with a backbone model. There were four datasets used to test and compare the proposed model to existing baseline approaches. The performance metrics (accuracy and dice score) for the proposed model showed marginal to moderate improvements over other high-performing models in classification and segmentation tasks. With that said, there is merit to this work and I'm generally satisfied with the results. Comments as follows:

1. The flow of the manuscript from Section 2 to Section 3 is odd. Consider moving Section 3 (Related Works) after the introduction.

2. I do not see any output(s)/inferences from the testing for CIFAR10/CIFAR100 datasets. A figure (with bounding boxes or masks) or even multiple figures showing how your model performed classification for these datasets is recommended. Same recommendation for when you use different backbones (Table 3).

3. What are examples of parameters mentioned in Table 4? Are there a subset of these params. that are important and if so, how do these affect performance?

4. Having equations for accuracy and dice score will help.

5. What are some limitations of your work? Since you've tested four different datasets and know what the data are like, you should be able to say something about in which cases (dataset type, size, classes, etc.) your proposed model may fall short.

6. Line 296 - Is this a typo or a placeholder?

Fair.

Author Response

(The authors gave the same response as above.)
